# Fast Anxiolytic-Like Effect Observed in the Rat Conditioned Defensive Burying Test, after a Single Oral Dose of Natural Protein Extract Products

**DOI:** 10.3390/nu13072445

**Published:** 2021-07-17

**Authors:** Thomas Freret, Stacy Largilliere, Gerald Nee, Melanie Coolzaet, Sophie Corvaisier, Michel Boulouard

**Affiliations:** 1UNICAEN, INSERM, COMETE, Cyceron, CHU Caen, Normandie University, 14000 Caen, France; stacy.largilliere@unicaen.fr (S.L.); gerald.nee@unicaen.fr (G.N.); coolzaet.melanie@gmail.com (M.C.); sophie.corvaisier@unicaen.fr (S.C.); michel.boulouard@unicaen.fr (M.B.); 2Behavioral Research Platform, Normandie University, 14000 Caen, France

**Keywords:** fish protein hydrolysate, anxiety, burying test, rodent

## Abstract

Anxiety appears among the most frequent psychiatric disorders. During recent years, a growing incidence of anxiety disorders can be attributed, at least in part, to the modification of our eating habits. To treat anxiety disorders, clinicians use benzodiazepines, which unfortunately display many side effects. Herein, the anxiolytic-like properties of two natural products (αS1–casein hydrolysate and Gabolysat^®^) were investigated in rats and compared to the efficacy of benzodiazepine (diazepam). Thus, the conditioned defensive burying test was performed after a unique oral dose of 15 mg/kg, at two time-points (60 min and then 30 min post oral gavage) to show potential fast-onset of anxiolytic effect. Both natural products proved to be as efficient as diazepam to reduce the time rats spent burying the probe (anxiety level). Additionally, when investigated as early as 30 min post oral gavage, Gabolysat^®^ also revealed a fast-anxiolytic activity. To date, identification of bioactive peptide, as well as how they interact with the gut–brain axis to sustain such anxiolytic effect, still remains poorly understood. Regardless, this observational investigation argues for the consideration of natural compounds in care pathway.

## 1. Introduction

Anxiety disorders are among the most prevalent and disabling psychiatric disorders worldwide [1]. Currently, on the market anxiolytic drugs are mostly benzodiazepine or benzodiazepine-like agents, which particularly target the GABAergic system. Unfortunately, their use is not harmless and requires caution and vigilance. Indeed, a life-threatening anxiety rebound effect may appear when withdrawal of benzodiazepine is too rapid [2,3,4]. Furthermore, benzodiazepines display significant side effects and drug interactions [5]. Risk-taking behavior have been observed under benzodiazepines prescription [6,7], together with impaired cognition, mobility and driving skills, as well as increased fall risk [8]. Additionally, if their consumption lasts for a long period (usually considered as 3 months), GABAergic medications might create a tolerance and dependence [5]. Finally, remote adverse effects have more recently been suspected, notably with an increased risk, to develop an Alzheimer’s disease [9].

In view of the risks associated with benzodiazepine intake, intense preclinical and clinical research were carried out to identify/develop new therapeutic strategies for the management of anxiety disorders. Therefore, anxiolytic drugs free of GABAergic action (such as buspirone) or even anti-depressant drugs (such as Selective Serotonin/Norepinephrine Reuptake Inhibitors) have been proposed as a substitute. Although they might be used routinely to cure general anxiolytic disorders, they are unfortunately devoid of fast-acting effects. In addition, they may lack therapeutic efficacy and/or display important side effects. Thus, apart from only a few drug developments and/or repurposing of some antidepressant drugs, fast-acting, efficient and safe therapeutic alternatives are scarce [10].

Surprisingly, hope for a new therapeutic strategy could come from diet supplements. Indeed, they are increasingly considered as a natural alternative to treat—or to manage—anxiety disorders [11]. This assumption comes, at least in part, from an observed close relationship (that some would qualify as a causal relationship) between changes in eating habits (mostly in Western countries) and the occurrence of anxiety disorders (associated or not with sleep disturbances) [12]. As a consequence, increasing efforts have been made in the search for natural, non-chemical, diet supplements with anxiolytic-like effect. Notably, natural peptides extracted from animal proteins appear as promising candidates. Thus, several interesting results were reported from experiments conducted with the milk-derived αS1–casein hydrolysate or with the fish protein hydrolysate. In fact, under a chronic oral administration regimen, both hydrolysates demonstrated anxiolytic-type properties in rodents (both mice and rats [13,14,15,16,17]), but also in several other animal species such as cats and dogs [18,19,20,21]. A few investigating works in humans have also been published ([22,23], see also for review [24]). For instance, level of state anxiety—assessed by the STAI test—was reduced in young adult students aged 18 to 25 after 1 week of supplementation with fish protein hydrolysate [25]. Similarly, 30 days of daily administration of αS1–casein hydrolysate (150 mg) appeared efficient to relieve 63 women suffering from stress-related symptoms [23]. Of note, treatment efficiency was in this work evaluated through a newly constructed questionnaire on a mix of two previously published ones (Hamilton Anxiety scale and Ferreri Anxiety Rating Diagram). Nevertheless, other stress-related physiological parameters seem to benefit from oral intake regimen of hydrolysates. Indeed, a dampening effect in stress-induced variations was also observed on different physiological parameters, such as stress hormone level (corticosterone) or systolic blood pressure [22].

To further examine underlying mechanisms, several preclinical investigations have been performed and each reported an anxiolytic effect either with one or the other natural hydrolysates. To date, no study has compared similar experimental protocols for the onset/efficacy of different hydrolysates. In fact, as worthwhile it can be, such a comparison is not possible given the high level of disparities in terms of tested doses or administration regimen as well as behavioral test used. For instance, anxiolytic properties of fish protein hydrolysate were reported in the elevated plus maze and in the defensive burying test, as well as in the conditioned light extinction test. All those experiments were conducted using a chronic administration regimen, i.e., an oral gavage twice a day, for either 3 or 8 days, with doses ranging from 25 to 100 mg/kg [15,26]. In addition, when using even higher range doses (i.e., 300 and 1200 mg/kg during 5 days), biochemical experiments demonstrated that Gabolysat^®^, a proprietary fish protein hydrolysate, displayed a diazepam-like effect on stress responsiveness of the pituitary–adrenal system and sympathoadrenal activity (with a reduced adrenaline and noradrenaline level in a stress condition and increase brain hippocampal GABA content in a non-stress condition) [13]. Conversely, when anxiolytic properties of αS1–casein hydrolysate were investigated under an acute administration regimen, a dose effect study (from 5 to 50 mg/kg) demonstrated that the minimal effective dose (15 mg/kg) was as efficient as diazepam (3 mg/kg) in the conditioned burying test. This beneficial effect was then observed for 60 min after a single oral administration [16,17].

## 2. Materials and Methods

We aimed to assess and compare acute anxiolytic properties (i.e., after a single oral dose) of two compounds (Gabolysat^®^ and αS1–casein hydrolysate), through a one trial burying behavioral test. This test was chosen given its high degree of face and construct validity, but also given its good pharmacological validation (predictive validity) [27]. Efficacy of the tested compound was compared to a reference drug (the most frequently consumed benzodiazepine on the market, i.e., diazepam).

Animals: All experiments were carried out in accordance with the European Communities Council Directive (63/2010) regarding the care and use of animals for experimental procedures, and they were approved by the local ethics committee (Comité d’Ethique NOrmandie en Matière d’EXpérimentation Animale, CENOMEXA, agreement number: 03-08-11/16/08-14). Aged of 8 weeks, male Wistar RjHan:WI rats were obtained from Janvier Labs (France) and pair-housed in Plexiglas cages with ad libitum access to food and water. A total number of 102 rats were necessary to perform displayed experiments. The animal facility was maintained on a controlled light–dark cycle (lights on from 7 a.m. to 7 p.m.), with a constant temperature (22 ± 2 °C). After 1 week acclimatization, animals were daily handled (2 × 5 min) during the following week to reduce the stress induced through the restraint required for oral gavage (Figure 1).

Group testing: αS1–casein hydrolysate and Gabolysat^®^ were obtained from Ingredia^®^ (Aras, France) and Laboratoire Dielen^®^ (Cherbourg, France), respectively. Diazepam (Sigma, France) was used as the pharmacological benzodiazepine references substance. All compounds were solubilized in a saline (NaCl, 0.9%) mix with 1% Carboxymethylcellulose (Sigma Aldrich^®^), which was used as treatment for the control group. The dose of 15 mg/kg was selected according to previous related studies [13,17]. Each compound was orally administered in a volume of 2 mL/kg body weight.

Defensive probe-burying test: Animals were familiarized in pairs for 2 consecutive days (20 min session) with the testing room (dim lit) and apparatus which consisted in a Plexiglas acrylic cage 42.5 × 26.6 × 18.5 cm (Intelli-bio^®^, Seichamps, France) [28,29]. The floor of the cage was covered with 5 cm of bedding material (fine wood sawdust). The following day, each rat was individually placed in the testing cage, with an electrified probe (7 cm long) emerging from one of its walls 2 cm above the bedding material. The latency for first approach to the probe, reflecting level of exploration behavior, was then recorded to ensure no bias interpretation of the data. Once the rat touched the probe, it received an electric shock of 0.2 mA (constant current shocker). All tested rats, whatever group or experiments considered, touched the probe only once and, therefore, received only a single shock. Burying behavior (pushing the sawdust ahead with rapid alternating movements of the forepaws oriented to cover the electrified probe) has been directly related with the experimental anxiety levels [29]. This behavior was constantly recorded during a 15 min period and in 5 min time slices.

Statistical analysis: All graphs displayed results as the mean ± standard errors of mean (SEM). Statistical analyses were performed with Statview^®^. ANOVAs were performed, followed when appropriate by post-hoc group comparison tests. The *p*-value was set at 0.05. Bonferroni correction was applied for post-hoc multiple comparisons.

## 3. Results

### 3.1. Evaluation of Anxiolytic-Like Effect (60 min after Oral Gavage)

One-way ANOVA of elapsed time to first approach to the probe (probe contact latency) did not reveal any statistical group differences (F_(3,48)_ = 0.301, *p* = 0.8245), thus, ensuring no locomotor and/or exploration behavior bias (Figure 2A). Furthermore, two-way ANOVA with a repeated measurement of burying time during the behavioral test did not reveal a group effect (F_(3,48)_ = 1.655, *p* = 0.1892), but a significant time effect (F_(2,96)_ = 27.773, *p* < 0.0001) and a group x time interaction (F_(6,96)_ = 3.158, *p* = 0.0072) (Figure 2B). When focused on the first 5 min of the test, a significant group difference was revealed (one-way ANOVA, F_(3,48)_ = 5.390 and *p* = 0.0028). In fact, the post-hoc test showed that all treated groups of rats (Diazepam, αS1–casein hydrolysate as well as Gabolysat^®^-treated animals) spent significantly less time burying the probe compared to the control group (*p* = 0.0162, 0.0159 and 0.0362, respectively). Thereafter (for the last 10 min of the test), probe-burying time for the control group massively dropped down (time effect: F_(2,24)_ = 52.373 and *p* < 0.0001). No group difference can then be revealed for the last 10 min of the test (*p* > 0.05).

### 3.2. Evaluation of Rapid (30 min after Oral Gavage) Anxiolytic-Like Effect 

One-way ANOVA of elapsed time to first approach to the probe did not reveal any statistical group difference (F_(3,46)_ = 0.376, *p* = 0.7705), thus, ensuring no locomotor biases (Figure 2C). Furthermore, two-way ANOVA with a repeated measurement of burying time during the behavioral test revealed a significant group (F_(3,46)_ = 12.749, *p* < 0.0001) and time effect (F_(2,92)_ = 75.708, *p* < 0.0001), as well as a group x time interaction (F_(6,92)_ = 3.779, *p* = 0.0021) (Figure 2D). Whichever group considered, animals spent less and less time burying the probe during the behavioral test (Control group: F_(2,20)_ = 53.993, *p* < 0.0001 and for diazepam, αS1–casein hydrolysate and Gabolysat^®^-treated group: F_(2,24)_ = 7.490, 25.232 and 12.089, respectively with *p* < 0.001). The post-hoc tests showed that the global group difference relied on significantly lesser probe burying time for diazepam- and Gabolysat^®^-treated animals compared to the control group (*p* = 0.0001 and 0.0005, respectively). In addition, if we consider only the first or second 5 min-section of the test, one-way ANOVA of burying time revealed a significant group difference (F_(3,46)_ = 12.981 and 4.556, with *p* < 0.0001 and *p* = 0.0071, respectively). The post-hoc tests showed that both diazepam- and Gabolysat^®^-treated animals spent significantly less time burying the probe compared to the control group (from 0 to 5 min test: *p* < 0.0001 and *p* = 0.0002; for 5–10 min test: *p* = 0.0150 and 0.0463, respectively).

## 4. Discussion

We herein experimentally demonstrated acute anxiolytic-like properties of Gabolysat^®^ and αS1–casein hydrolysate [17]. Orally given at a dose of 15 mg/kg and tested in the conditioned burying task 60 min later, both natural products were as efficient as diazepam (used as benzodiazepine reference pharmacological drug) to elicit anxiolytic activity. More interestingly, when the anxiolytic-effect was assessed even as soon as 30 min after the oral dose, a similar efficiency (still compared to diazepam) was observed for Gabolysat^®^. This last result demonstrates a fast onset of anxiolytic activity for Gabolysat^®^.

Diazepam is a well-known and frequently prescribed benzodiazepine, which already experimentally demonstrated its anxiolytic activity in the conditioned burying test. Thus, its anxiolytic activity was described either when administrated intra-peritoneal (doses ranging from 0.5 to 2 mg/kg) [29,30], or orally given (3 mg/kg) [16,17]. Despite any potential differences of laboratory experimental conditions (light/dark cycle, strain, testing room environment, …), an anxiolytic effect of diazepam is always found, as herein. The same holds true for αS1–casein hydrolysate. Indeed, when administered orally at 15 mg/kg and then tested 60 min after, αS1–casein hydrolysate demonstrated anxiolytic properties in the conditioned burying test. Such a result is in accordance with literature data [16,17], reinforcing our choice to use αS1–casein hydrolysate as a reference natural product having demonstrated anxiolytic activity. Here, we demonstrated for the first time, that Gabolysat^®^—a fish proteins extract—also has rapid anxiolytic properties, as testified by the reduced probe-burying time observed in animals with acute treatment. Additionally, one can note that, whichever treatment was considered, the animals’ anxiety levels (behaviorally assessed through probe-burying time measurement) were similar (no statistical difference between treated animals’ groups). Thus, within the sensitivity limits of the test, we observed similar anxiolytic properties of each of the natural compounds and the benzodiazepine.

In a rat, the maximum plasmatic concentration of diazepam *per os* is observed at 30 min [31]. Hence, the experiment was conducted again with a shorten elapsed time (30 min) after oral dose to attempt to observe a rapid anxiolytic effect of both natural compounds and benzodiazepine. Since the maximum plasma concentration of diazepam is observed 30 min after oral gavage, this was taken as the new elapsed time to evaluate anxiolytic properties. Thus, we demonstrated for the first time to the best of our knowledge, that diazepam—but also Gabolysat^®^—have rapid anxiolytic properties. This was testified by the decreased probe-burying behavior during the first 10 min of the test, performed only 30 min after oral dose. This last finding regarding the fast onset of Gabolysat^®^ could draw clinicians’ attention as it might find application in several disorders (such as sleep anxiety and insomnia) or even psychiatric pathologies, such as depression. Indeed, depression, for instance, displays a 57% rate to co-occur with one anxiety disorder. Antidepressants prescribed so far, have revealed to be moderately efficient. This is firstly because some patients do not respond to treatment, and then because the average time to achieve remission for those who are sensitive to it is approximately 7 weeks [32]. Thus, natural products such as Gabolysat^®^—provided they are devoid of toxicity—may offer a therapeutic adjuvant during the first phase of treatment. So far, no side effect was reported in rodents after chronic (5 days) oral treatment of high doses of Gabolysat^®^ (dose ranging from 300 to 1200 mg/kg) [13].

Beyond toxicity, the mechanisms underlying anxiolytic activity observed after an oral dose of those two natural compounds still remains unknown. Given the different behavioral profile observed in our study, one might imagine that their mechanisms may slightly diverge. Among possible mechanisms, a direct action on central GABAergic and/or serotonergic transmission, as well as modulatory role on corticotrope axis have shown preliminary results [13,33]. However, one exciting hypothesis recently emerged from current ongoing research on microbiota–gut–brain axis functioning. Indeed, gut peptides were recently evoked as important regulators of microbiota–gut–brain signaling in health and stress-related psychiatric illnesses [34]. Further works are required to investigate how those two natural compounds might interact with microbiome and, as a consequence, on gut–brain axis functioning.

A major study limitation is the chosen route of administration. In fact, while closely related to human practice, oral gavage required animal restraint which is a stress-induced event. In addition, the technique might be harmful or induce pain (esophageal trauma, etc.), if the experimenter is not skilled in animal handling and restraint. All animals herein underwent handling/restraint by the same experimenter (with acknowledged expertise) and the animals’ groups were counterbalanced across the session.

## 5. Conclusions

In conclusion, our results confirm the efficacy of αS1–casein hydrolysate to dampen anxiety level but they also demonstrate, for the first time, the efficacity of Gabolysat^®^ to do so. In addition, this last compound also displays a fast onset of action affording a rapid and lasting anxiolytic action. Numerous natural products are marketed for behavior therapy, but very few have demonstrated any evidence of efficacy. Additionally, as both natural products were described as being devoid of any major side effects, they appear as powerful natural alternative solutions to benzodiazepine drugs and to their constraints of use.

## Figures and Tables

**Figure 1 nutrients-13-02445-f001:**
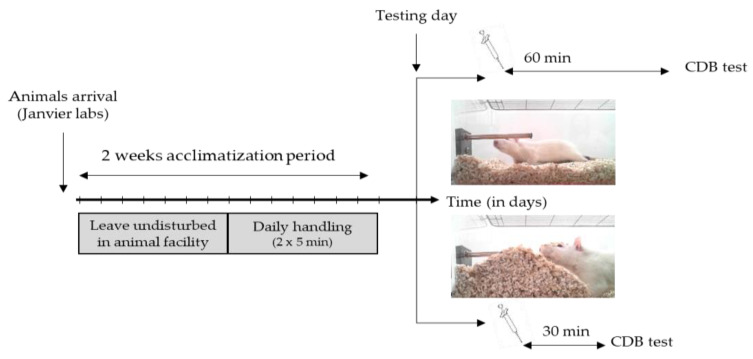
Experimental design. There was a 2 week acclimatization period, where animals were first left undisturbed for the first week, then handled twice daily in order to acclimate the animals to the experimenter. On the testing day, 60 or 30 min after an oral gavage, the Conditioned-Defense Burying (CDB) task was performed.

**Figure 2 nutrients-13-02445-f002:**
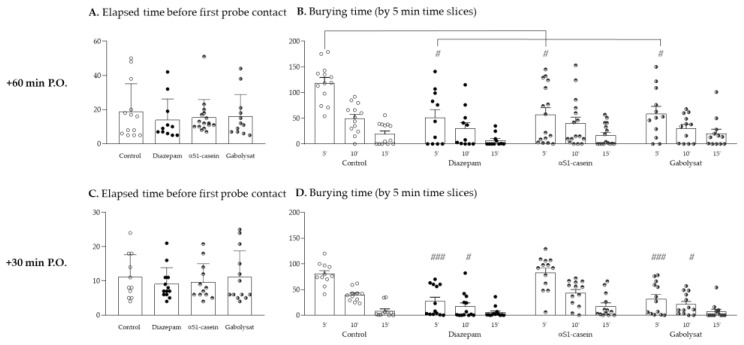
Conditioned behavioral test performed 60 min (**A**,**B**) and 30 min (**C**,**D**) after oral gavage. Figure 1 displays the latency for first contact to probe, as a control measurement of motivation/locomotor activity for all animals’ groups. Figure 1 display probe-burying behavior during the test (by 5 min time slices), reflective of anxiety levels. A total of 102 rats were used. Groups sizes were 13, 11, 16 and 12 for control, diazepam, αS1–casein hydrolysate and Gabolysat^®^, respectively, when the test was performed 60 min after gavage (**A**,**B**); corresponding 13, 11, 13 and 13 when test was performed 30 min after gavage (**C**,**D**) (ANOVA with repeated measurement, post-hoc test: # *p* < 0.05 and ### *p* < 0.001 compared to respective control time).

## Data Availability

The data presented in this study are available on request from the corresponding author.

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
