# Peer review of "Fast Anxiolytic-Like Effect Observed in the Rat Conditioned Defensive Burying Test, after a Single Oral Dose of Natural Protein Extract Products"

_nutrients, 2021, doi:10.3390/nu13072445_

Round 1

Reviewer 1 Report

Freret et al have investigated the anxiolytic properties of two natural compounds, αs1-casein hydrolysate and fish protein hydrolysate and compared it with a benzodiazepine, a drug that was the first line of defence for anxiety treatment but is now primarily used in crisis due to serious issues with tolerance, withdrawal and other adverse side effects. The need to find new anxiolytics that are as effective but with fewer side effects is paramount. I therefore find this study highly topical and their finding that both compounds could offer acute relief to an anxiety-driven behaviour is promising as an initial finding. I do however have some comments that need addressing:

Although both compounds display anxiolytic properties that appear similar to diazepam, this finding is limited by the single dose tested and the lack of any side effect studied. It would have been more informative if a dose-effect profile was conducted to ascertain useful pharmacological information such as ED50 and compare this with any side effects across the dose ranges and thus determine the therapeutic index. It is well known diazepam causes sedation and other side effects. The authors anecdotally report no obvious side-effects with these compounds and reference a single study that used very different treatment regimen for Gabolysat. Therefore, no formal testing of side effects compared with control was conducted. In my opinion, this is a severe limitation to this study.

There is a significant degree of variation in burying time in the groups that received diazepam, αs1-casein and Gabolysat. In fact, the single data points indicate a divide in the data where some rats were responsive to the treatment intervention while others remained insensitive.  This is more obvious at 60 minutes post gavage (5 minute time epoch) but can also be seen in Diazepam and Gabolysat at 30 minutes post gavage. This was not mentioned in the discussion. Can the authors comment on the reason for this?

The use of ‘kinetics’ throughout the manuscript is misleading as no formal kinetic analysis of the drugs were conducted and therefore should be removed.

It is not clear how ‘time burying’ was quantified within the 5 minute time epochs, this needs to be clarified in the methods.

Non of the graphs have y-axis labels, what are the units here?

Figure 2D, what do the #’s signify i.e. from which condition do they signify a significant difference? Please clarify this either on the graph or in the legend.

Page 2, line 45 should be therapeutic efficacy not efficiency.

Page 2 lines 66-67: Since proteins are extremely diverse and can perform a plethora of different functions, I do not think the common ‘protein’ nature of the two compounds is enough reason to suggest their actions rely on a common mechanism of action! This sentence needs rethinking.

There are a number of instances throughout the manuscript where the language is a difficult to read and could benefit from some grammatical/sentence restructuring. For example, page 7 lines 226-229.

Reviewer 2 Report

First of all I would like to congratulate the authors for their work.

Line 61.- It is not explained whether the reduction in anxiety in that study 25 (Dorman, TB, L .; Glaze, P .; Hogan, J .; Skinner, R .; Nelson, D .; Bowker, L .; Head, D.. The effectiveness of Garum armoricum (Stabilium) in reducing anxiety in college students. Journal of Advancement in Medicine 1995 ,8, 193-200, doi: 10.1001 / jama.2011.982) occurs in both types of anxiety (trait and state or not and the possible explanations in this regard

References 22, 23 (line 64). It is said of referred stress symptoms, measured through questionnaires. Which ones should be specified, because otherwise the statement lacks rigor.

In the use of these references some methodological problems are observed:

-Reference 22. The Stroop test that measures attention is used, and there is no adequate justification for its use to relate it to anxiety, which is the objective of using it as a reference. Blood pressure is used as a measure of stress level, which is not necessarily a determinant of anxiety, since it can have an idiopathic or essential etiology.

-Reference 23. Uses a questionnaire to measure stress that has not been validated, which makes the results less valid. The authors should better justify the use of these two texts with regard to the measurement of anxiety.

-Lines 86-90. This text corresponds to the method section.

Method: unequivocally specify the study design (type of study and its justification) at the beginning of the section.

These questions should be answered by the authors clearly in the discussion:

-Do the authors believe that these results can be extrapolated to humans? That is, what are the differential factors that can act as biases in the attribution of the results?

-Can laboratory conditions influence the results found? In what sense?

-Do the authors believe that the reduction in anxiety affects both dimensions of the construct (trait / state) that the STAI questionnaire measures?

Author Response

This manuscript is a resubmission of an earlier submission. The following is a list of the peer review reports and author responses from that submission.

Round 1

Reviewer 1 Report

Freret et al have used the behavioural paradigm defensive probe burying as an assay to assess the anxiolytic properties of two natural products α-casozepine and Gaboysat against diazepam. They found rats treated with both compounds displayed less anxious behaviour, equivalent to those treated with benzodiazepines. Further, whilst α-casozepine did not appear to have an effect until 1hr after administration, those treated with Gaboysat, displayed anxiolytic behaviour 30 mins after administration.

Although this work has some merit, I do not think it is suitable for publication in its present form. Even as a brief report, I find the findings are somewhat limited and could be expanded upon. For example: use of more than one behavioural paradigm to assess anxiety &/or side effects; assessing the dose-response of the compounds (+ side effects); attempt to find a mechanism of action (e.g. GABAergic?). Any one of these would increase the significance of their findings. In addition, this manuscript needs major re-writing. The introduction and discussion make no attempt to give any detail on the background of these two compounds, their anxiolytic properties or their possible mechanism of action. Thus, the justification for this study – as presented by the authors in their introduction – is lacking. A mention was given to unknown side effects of natural compounds, but this was not assessed in the current study. It is also alluded that α-casozepine has been shown to be anxiolytic before, but no details are given. In addition, the authors claim that this is the first demonstration of Gaboysat having anxiolytic properties however, this is not true. Bernet et al (2000) (PMID: 10789880) have already demonstrated Gaboysat has anxiolytic properties by reducing the stress-responsiveness of the HPA-axis, similar to diazepam. Further, a study by Landsberg et al (2015) J. Veterinary Behavior show a similar fish hydrolysate is anxiolytic in dogs. The authors need to make clear what their study adds to this existing literature.

Although shock-probe burying is a validated method to assess anxiety levels in a rodent, it can be difficult to interpret whether burying behaviour vs immobility or increased time to bury reflects differences in anxiety levels or different coping styles (defensive vs passive). Did the authors also measure immobility/freezing time?

More detail is needed in the methods:

What doses of each compound were administered – it is stated that rats received 2ml/kg but this means nothing without the dose or concentration of the drug.

Was the probe electrified during the entire 15 minutes of testing? i.e. Did the rats receive a single shock on first approach of the probe or was it shocked each time it touched the probe during the whole 15 minutes?

What posthoc tests were used?

Reviewer 2 Report

The authors have done a very good job in conducting the experiment and preparing the manuscript. Still, there are a few aspects that need to be addressed. First of all, although I do not feel qualified to judge the English language, I strongly recommend proofreading the manuscript (starting with the abstract). At times it’s hard to read it and some phrases are ambiguous.

The Title should be more specific. The experiment targeted only two natural protein extract products (and Diazepam) and not bioactive peptides in general. Also, in my opinion, the title suggests a theoretical approach and not an experiment.

The Abstract gives a good account of the content of the manuscript, but needs to be more detailed concerning the results.

Keywords. The authors tested two natural protein extract products (from either fish or cow milk), not to mention Diazepam. Also, stress? Maybe anxiety would be more appropriate, given the purpose of the experiment (later in the manuscript, the authors stated that “Burying behavior (pushing the sawdust ahead with rapid alternating movements of the forepaws oriented to cover the electrified prod) has been directly related with the experimental anxiety levels”).

The Introduction section should provide a little more information and needs to be clearer. The first paragraph is too long and hard to follow. Maybe it would be a good idea to separate it into two paragraphs. Also, I would like some references for two phrases: “Thus, overuse of benzodiazepine drugs is a growing problem in clinical practice, notably in France” and “Benzodiazepine use remains too high, especially among the elders, who are their highest consumers”. Also, please provide references for the affirmations from lines 49 to 54 (and maybe some details regarding the changes in eating habits).

Furthermore, please clarify the reason for the inclusion of eating habits in the introduction (although they are not the target of the experiment). Why does the introduction only refer to humans if the experiment is performed on animals?

Materials and method

Animals. How many? The authors give a detailed account of the number of rats involved in each experimental condition when describing Figure 2, but I would like them to add the total number.

Statistical analysis. “All results are expressed as the mean±SEM”? Where? Also, the authors listed only two-way ANOVA with repeated measurements, although they also presented the results of one-way ANOVAs. “P-value was set at 0.05 and adjusted according for multiple post-hoc comparisons” – what do you mean by adjusted according?

Defensive prod-burying test. When did the rats receive the substances? From the description and Figure 1, I understand that, on the testing day, the rats received the substances (according to each experimental condition), then placed in the cage with the buried prod. After being placed in the cage, latency for first approach to the probe was recorded and, once the rat touched the prod, it received an electric shock and its behavior was constantly recorded during a 15min period.

Results

I would like to point out that there is a redundancy in reporting the results of the subsection Evaluation of anxiolytic-like effect (60min after oral gavage). The authors used two-way ANOVA with repeated measurements in order to test the group and time effects and their interaction on burying time, then reported the results of one-way ANOVA, and continued with reporting the results of post-hoc tests. The authors were clearer when reporting the results for the Evaluation of rapid (30min after oral gavage) anxiolytic-like effect.

Figures. Please provide a description for Figure 1 (along with what CDB stands for). When describing Figure 2, the authors mentioned “Fig1A and C” and “Fig 1B and D”, but there are no Fig 1A and 1B.

Discussion

Please include some limitations of the study (for example, those associated with oral gavage, such as esophageal trauma and aspiration).

“So far, we observed no side effect in rodents after chronic (5days) oral treatment of high doses of Gabolysat® (dose ranging from 300 to 1200mg/kg) [15]”. The word "we" gives the impression that the results come from an experiment performed by the authors of the present manuscript.

Reviewer 3 Report

The authors present a short manuscript of anxiolytic effects of Casozepine or Gabolysat acute administration compared to those of diazepam in young adult male Wistar Rats.

The manuscript is well written, the aims of the study clearly presented. Methods and results are clear. However it seems very preliminary.

I have however some concerns:

1) How authors explain the variation in latencies or on burying time between control group in the 30 or 60 min experiment. 

2) Symbols on histogram bars, for instance in graph 1B, suggest a p<0.001 vs the control group, which is not what is described in the text.

3) Because of the histogram bars representation, it is difficult to observe the groups distribution. 

4)Effects in the Defensive prod-burying test should be confirmed in another behavioral procedure, such as EPM to be more convincing.

5) A greater description of  Casozepine or Gabolysat properties or prior studies in the introduction would be appreciated.

Reviewer 4 Report

The manuscript by Freret et al. details anxiolytic-like effects of two commercial protein protein extracts (one from fish and one from bovine).  The authors show that these two commercial protein extracts have equivalent or even better anxiolytic effectives than benzodiazepine.  There is no question that benzodiazepine is far from a perfect drug and alternatives are needed to replace benzodiazepine as a treatment for anxiety disorders, but this work is, at best, a very modest addition to the field.

  1. No justification of using only male rats in these studies.  Does the use of only male rats have any impact on the conclusions drawn by the authors.
  2. The use of relatively crude mixtures of protein extracts renders this work of limited scope and use. Even if these data are completely valid, it would be far more useful to identify specific components from the protein extracts that exhibit anxiolytic effects.
  3. The use of protein extracts is inconsistent with the title of the manuscript. The term “bioactive peptides” is mis-leading.
  4. The authors have not included a control that might demonstrate that the results in this manuscript result from a simple feeding effect. Would the same results have been observed if the rats were given oral gavage of some other non-protein caloric source?